# Biosignal Compression Toolbox for Digital Biomarker Discovery

**DOI:** 10.3390/s21020516

**Published:** 2021-01-13

**Authors:** Brinnae Bent, Baiying Lu, Juseong Kim, Jessilyn P. Dunn

**Affiliations:** 1Department of Biomedical Engineering, Duke University, Durham, NC 27708, USA; brinnae.bent@duke.edu (B.B.); baiying.lu@duke.edu (B.L.); juseong.kim@duke.edu (J.K.); 2Department of Biostatistics and Bioinformatics, Duke University, Durham, NC 27708, USA

**Keywords:** wearables, biosignal, data compression, data compression algorithms, data, electrocardiogram, photoplethysmography, accelerometry, electrodermal activity

## Abstract

A critical challenge to using longitudinal wearable sensor biosignal data for healthcare applications and digital biomarker development is the exacerbation of the healthcare “data deluge,” leading to new data storage and organization challenges and costs. Data aggregation, sampling rate minimization, and effective data compression are all methods for consolidating wearable sensor data to reduce data volumes. There has been limited research on appropriate, effective, and efficient data compression methods for biosignal data. Here, we examine the application of different data compression pipelines built using combinations of algorithmic- and encoding-based methods to biosignal data from wearable sensors and explore how these implementations affect data recoverability and storage footprint. Algorithmic methods tested include singular value decomposition, the discrete cosine transform, and the biorthogonal discrete wavelet transform. Encoding methods tested include run-length encoding and Huffman encoding. We apply these methods to common wearable sensor data, including electrocardiogram (ECG), photoplethysmography (PPG), accelerometry, electrodermal activity (EDA), and skin temperature measurements. Of the methods examined in this study and in line with the characteristics of the different data types, we recommend direct data compression with Huffman encoding for ECG, and PPG, singular value decomposition with Huffman encoding for EDA and accelerometry, and the biorthogonal discrete wavelet transform with Huffman encoding for skin temperature to maximize data recoverability after compression. We also report the best methods for maximizing the compression ratio. Finally, we develop and document open-source code and data for each compression method tested here, which can be accessed through the Digital Biomarker Discovery Pipeline as the “Biosignal Data Compression Toolbox,” an open-source, accessible software platform for compressing biosignal data.

## 1. Introduction

Wearable sensors have the potential to transform health management and healthcare delivery. However, the digital footprint of this biosignal data is growing at an unprecedented rate; the number of connected wearable devices worldwide is expected to reach 1.1 billion by 2022 [1]. Immense data storage capacity is necessary to retain information collected from wearable sensors that continuously monitor multiple biosignals. This “data deluge,” combined with the high costs of data storage and challenges associated with efficient data organization [2,3,4], reveals a critical need to determine how to reduce biosignal data volumes appropriately to retain important information while removing unnecessary or repetitive information [5].

Digital biomarkers are digitally collected data (e.g., a heart rate biosignal from a wrist-worn wearable) that are transformed into indicators of health outcomes (e.g., risk of cardiovascular disease). Digital biomarkers have applications in a number of disease states, including movement-related disorders [6], breast cancer [7], and Alzheimer’s disease [8]. They are currently being used to diagnose and monitor a number of chronic diseases and conditions [9], and the field of digital biomarkers is rapidly expanding [10]. In order to develop robust digital biomarkers, high-resolution biosignal data are often necessary. Thus, data volumes must be reduced in such a way that the information necessary for digital biomarker development is retained. Hence, the importance of robust data compression methods that have been tested on biosignal data is critical for developing digital biomarkers while minimizing data footprints. There have been multiple methods explored to consolidate biosignals from wearables. Data aggregation in the form of daily summaries rather than high-resolution data is widely used by commercial wearable manufacturers, but these aggregated metrics are often insufficient for the development of digital biomarkers for healthcare applications. The use of downsampling for biomedical signal compression has been used for decades and has recently been examined for digital biomarker development [5,11]. Recent studies have provided the minimum sampling rates necessary for digital biomarker development in several types of wearable sensors [5,12], highlighting the limits of downsampling and the need for data compression.

Data compression refers to the methods used to store or send data in a smaller number of bytes than the original representation [13]. There is a tradeoff between compression of a signal (compression ratio) and minimizing information loss [14]. Data compression has been explored for biomedical signals previously, but the evaluation of these methods has been limited to one type of sensor, including either electrocardiogram (ECG) [14,15,16,17], photoplethysmography (PPG) [18,19,20], accelerometry [21,22], and electrodermal activity (EDA) [23]. Data compression methods can be divided into two groups: Lossless compression and lossy compression. Here, we focus on lossless compression methods, which are anticipated to be most useful for digital biomarker development due to them allowing the data to be fully reconstructed [24,25].

There are two primary characteristics of signals that affect the performance of data compression methods: Redundancy in the signal and stationarity of the signal [26]. While biosignals are inherently nonstationary, the duration and extent of nonstationarity varies for different biosignal types. This helps to determine which compression method is optimal for each biosignal type. Redundancy of measurement values also varies widely as a result of differences in variations of biosignals over time. For example, there is a low redundancy in accelerometry signals due to continual movements, while skin temperature signals remain consistent for longer stretches of time and therefore has high redundancy. In addition to differences in redundancy, many biosignals are periodic, including ECG, PPG, and accelerometry. Compression methods utilizing wavelet-based transforms have demonstrated high performance on periodic signals [25]. A comprehensive analysis of frequently used data compression methods applied to biosignals from common wearable sensors would uncover how data compressibility and recoverability are affected by the characteristics of biosignals. This research could point to new directions for innovation in wearable sensor data compression methodology.

Transform domain data compression can generally be divided into two steps: Transformation and encoding. Based on a literature review for data compression methods that have reported high performance and preservation of information in data with similar characteristics to wearable sensor biosignal data, including high redundancies, nonstationarity, and periodic behavior, we have implemented three separate transformation methods in this study. These methods include: Singular Value Decomposition (SVD), Biorthogonal Discrete Wavelet Transformation (BD-WT), and Discrete Cosine Transformation-based Discrete Orthogonal Stockwell Transformation (DCT-DOST) in addition to a direct, encoding-only method, which we refer to as Direct Data Compression (DDC). SVD is used to factorize the signal into three smaller sets of values [27], which preserves features for digital biomarker development. BD-WT is widely used in ECG data compression [14,16,23,28]. DCT-DOST is a more recent method that has shown to be very robust for ECG signal compression [25]. For data encoding, we implemented two separate methods: Huffman encoding (HE) and run-length encoding (RLE). HE is a lossless encoding method that has been reported as an efficient coding method for wearable sensors [29]. RLE exploits the repetition of data observation values by representing the data as run and length [25], and can be applied alone or subsequent to the DCT-based DOST transformation method.

The primary objective of this study is to evaluate applying commonly used data compression methods including data compression algorithms (singular value decomposition, discrete cosine transform, and biorthogonal discrete wavelet transform) and encodings (run-length encoding and Huffman encoding) to common wearable sensor data, including ECG, PPG, accelerometry, EDA, and skin temperature (Figure 1). We have developed, documented, and released open-source code and data for each of these methods in the Digital Biomarker Discovery Pipeline [10] as the “Biosignal Data Compression Toolbox,” an open-source, accessible software platform for compressing wearable sensor biosignal data.

## 2. Materials and Methods

### 2.1. Data

The data used to evaluate the compression methods included ECG data from the Bittium Faros 180 sampled at 1000 Hz (Bittium Inc., Oulu, Finland) (original size = 330 MB) and PPG sampled at 64 Hz (original size = 34.4 MB), accelerometry sampled at 32 Hz (original size = 38 MB), EDA sampled at 4 Hz (original size = 4.4 MB), and skin temperature sampled at 4 Hz (original size = 2.2 MB) from the Empatica E4 (Empatica Inc., Milano, Italy) [30]. This equates to ~40 h of monitoring from the ECG and ~24 h of monitoring from each of the other sensors. The de-identified test data used for evaluation can be found in the Biosignal Data Compression Toolbox, available at DBDP.org.

### 2.2. Pre-Processing

We filtered the ECG signal through a band-pass filter (second-order Butterworth filter; low cut-off frequency = 0.5 Hz, high cut-off frequency = 50 Hz), resampled the signal to 200 Hz, and normalized the signal to the range [0, 1] following the standard procedure [27]. For the remaining wearable sensor data (PPG, accelerometry, EDA, and skin temperature), we only band-pass filtered the signal (second-order Butterworth filter; low cut-off frequency = 0.5 Hz, high cut-off frequency = 50 Hz).

### 2.3. Data Compression Evaluation Criteria

We evaluated the data compression pipelines using the compression evaluation metrics compression ratio (CR) [24] and percentage root-mean-square difference (PRD) [31,32]. CR is the ratio of uncompressed data size over compressed data size. Information loss is quantified through the percentage root-mean-square difference, and we therefore focus on minimizing the percentage root-mean-square difference between the original and reconstructed signal.
(1)$$CR=\;\frac{L_{original}}{L_{compressed}}$$

Let *x*[*n*] and $\hat{x}$[*n*] be the original (pre-processed) and the reconstructed signals, respectively, and *N* its length. μ[n] is the sample mean of the *N* data samples of the original signals.
(2)$$\;PRD=\;\sqrt{\frac{\sum_{n=1}^{N}{\left(x\left[n\right]-\;\hat{x}\left[n\right]\right)}^{2}\;}{\sum_{n=1}^{N}{\left(x\left[n\right]-\;\mu \left[n\right]\right)}^{2}}}\;\times \;100$$

### 2.4. Data Compression Methods

We implemented five data compression pipelines in this study, including: Direct Data Compression (DDC) with Huffman Encoding (HE), Singular Value Decomposition (SVD) with HE, Biorthogonal Discrete Wavelet Transformation (BD-WT) with HE, and Discrete Cosine Transformation-based Discrete Orthogonal Stockwell Transformation (DCT-DOST) with HE and with RLE, respectively. We describe each data compression pipeline briefly below.

#### 2.4.1. Direct Data Compression with Huffman Encoding

DDC with HE [28] encodes the pre-processed signal directly without any transformation. HE was performed directly on the data [33].

#### 2.4.2. Singular Value Decomposition with Huffman Encoding

To perform singular value decomposition (SVD), the sequence length was truncated to the closest square to generate an m × m matrix from the pre-processed data, which was then decomposed into U, Σ, and V matrices following the typical SVD procedure [27]. To compress the data, the rank of the matrix Σ is tuned by shrinking eigenvalues to zero until the PRD reaches 2% to produce Σ’, and then the original m × m matrix is reconstructed from U, Σ’, and V. After SVD, HE was applied.

#### 2.4.3. Biorthogonal Discrete Wavelet Transform with Huffman Encoding

BD-WT was implemented using a five-level wavelet decomposition on the pre-processed data with biorthogonal (bior4.4) as the wavelet base [14]. The energy of all of the coefficients was calculated, the threshold was computed for each coefficient matrix, and only the coefficients that were above the thresholds were kept. After filtering the data by energy, the data were quantized in steps, beginning at 8 bits and reducing by 1 bit for each iteration, in order to discover the optimal PRD. All coefficient arrays and binary maps separated by decomposition level were concatenated and then HE was applied.

#### 2.4.4. DCT-Based DOST with Huffman Encoding

The Discrete Cosine Transform (DCT)-based Discrete Orthogonal Stockwell Transform (DOST) [25] was implemented by first resampling the processed data into 2^N+1^ and then dividing the data into different groups: 1, 1, 2, 4, 8, …, 2^(N−1)^. Within each group, an inverse DCT was performed separately. The results from all groups were then stacked together into a whole array. After DCT-DOST, HE was applied.

#### 2.4.5. DCT-Based DOST with Run-Length Encoding

A DCT-based DOST was performed as described above [25]. On the approximated data, RLE was applied [25]. RLE is an effective lossless encoding method that encodes data by aligning the value and its count sequentially (e.g., “aaaaaaabbbbccc” is encoded as “a7b4c3”).

## 3. Results

We combined commonly used algorithmic data compression pipelines including direct data compression (DDC), singular value decomposition (SVD), the discrete cosine transform (DCT-DOST), and the biorthogonal discrete wavelet transform (BD-WT) with commonly used encoding data compression methods including run-length encoding (RLE) and Huffman encoding (HE) into five distinct data compression pipelines. We evaluated the compression ratio and percentage root-mean-square difference of each pipeline applied to common wearable sensor biosignal data, including ECG, PPG, accelerometry (ACC), EDA, and skin temperature (TEMP) (Figure 1). We have developed and documented open-source code for each compression pipeline tested here and contributed this and data for testing the methods to the Digital Biomarker Discovery Pipeline as the “Biosignal Data Compression Toolbox,” an open-source, accessible software platform for compressing biosignal data.

We explored the tradeoff between the PRD and CR across the five data compression pipelines applied to five different data types with the goal of minimizing the PRD while maximizing the CR (Figure 2). The CR ranged from 3.19 to 131.70 across the data compression pipelines. The PRD ranged from 0.02% to 10.67% across the data compression pipelines (Table 1). We demonstrate the tradeoff between compression of a signal (CR) and minimizing information loss (PRD) for each of the data compression pipelines and for each of the biosignals (Figure 3). Consistent with the literature, we find that higher compression ratios result in a higher PRD. However, we demonstrate that this tradeoff is not equal for all biosignals and all compression methods (Figure 3).

We demonstrated that different biosignals have varying performance with different compression methods, which may be dependent on signal characteristics, including redundancy and periodic behavior of the biosignals, which can be visualized in Figure 4. For example, because skin temperature is a relatively consistent signal with many redundancies (Figure 4), the tradeoff between CR and PRD was tempered; skin temperature maintained lower information loss while reducing data volumes as compared with the other biosignals. Furthermore, signals that were more similar to one another in terms of redundancies and periodic behavior (i.e., ECG and PPG) resulted in similar optimal data compression pipelines (both in terms of PRD and CR) (Figure 4).

### 3.1. Minimizing Information Loss for Digital Biomarker Development

For ECG and PPG, the compression pipeline that minimized the PRD was DDC (no transform) with HE (ECG PRD = 0.17%; PPG PRD = 0.06%). For EDA and ACC, the data compression pipeline with the lowest PRD was SVD with HE (EDA PRD = 0.14%; ACC PRD = 0.02%). For the skin temperature sensor (TEMP), both DDC (no transform) with HE and BD-WT with HE minimized the information loss (PRD = 0.09% and 0.10%, respectively). We present visualizations of the compression pipelines that minimize information loss most effectively for each biosignal in Figure 4 with the original signal, the reconstructed signal, and the difference between the original and reconstructed signals. This illustrates signal characteristics of each biosignal and the resulting differences between original and reconstructed signals, which are also represented in the PRD.

### 3.2. Maximizing Data Compression

While minimizing information loss is the most critical component of data compression for digital biomarker development, we also examined the data compression pipelines that maximized the compression ratio, which is most relevant when there are limitations to data volumes, for example, in Bluetooth-based data transfer. The data compression pipeline implemented in this study with the highest compression ratio for PPG, EDA, and ACC was DCT DOST with HE (PPG CR = 23.03; EDA CR = 19.64; ACC CR = 23.75). For ECG and TEMP, the data compression pipeline with the highest compression ratio was BD WT with HE (ECG CR = 131.70; TEMP CR = 78.90).

### 3.3. Open-Source Biosignal Data Compression Toolbox

The Biosignal Data Compression Toolbox is available open source in the Digital Biomarker Discovery Pipeline (DBDP). The DBDP is an open-source software resource [10] published in GitHub with Apache 2.0 licensing and is available at dbdp.org. Contribution to the Biosignal Data Compression Toolbox is encouraged following the Contributor Covenant (v2.0) [34] code of conduct. Contribution is encouraged following DBDP guidelines. As part of the Toolbox, we provide the test datasets used in this analysis: ECG (148.5 million measurements), PPG (5.3 million measurements), accelerometry (2.6 million measurements), electrodermal activity (330,000 measurements), and skin temperature (330,000 measurements).

## 4. Discussion

In this study, we demonstrated the feasibility, utility, and benefit of applying data compression tools to biosignals. We tested five common data compression pipelines on five different types of wearable sensor biosignal data: ECG, PPG, accelerometry, electrodermal activity, and skin temperature. Overall, we found that the most effective data compression pipeline varied by biosignal, and we report the optimal pipeline for compressing data from each wearable sensor type. This furthers our knowledge of how data compressibility and recoverability are affected by the characteristics of biosignals and points to new directions for innovation in wearable sensor data compression methodology. Finally, we have open-sourced our algorithms and data in the Biosignal Data Compression Toolbox as part of the Digital Biomarker Discovery Pipeline.

In order to develop robust digital biomarkers, high-resolution biosignal data are often necessary. However, the large volumes of longitudinal, high-resolution data required for digital biomarker discovery accrue large storage costs. Healthcare data storage requirements are quadrupling every 2–3 years and are projected to cost up to $600 billion/month in 2020 [3,5]. It is therefore critical to reduce biosignal data volumes. However, biosignal data must be preserved for digital biomarker discovery because digital biomarkers often require high precision. Thus, not only is it critical to reduce biosignal data volumes, but it is also necessary to ensure that information required for digital biomarker development is retained. The importance of robust data compression methods that have been tested on biosignal data is critical for developing digital biomarkers while minimizing data footprints.

Here, we evaluate the performance of each data compression pipeline using two metrics: The compression ratio (CR) and the percentage root-mean-square difference (PRD). The former tells us to what extent data volumes are reduced, while the latter tells us to what extent the information encoded in the original data is preserved. For digital biomarker development, we recommend evaluating this tradeoff for each biosignal and each application. For example, some digital biomarkers may require data with minimized information loss to be computed, such as the sensitive heart rate variability frequency domain metrics. In this case, minimizing the PRD is more of a priority than maximizing the CR because minimizing information loss is critical. Computing other digital biomarkers, such as skin temperature variability, may not require data that has minimized information loss. For these applications, maximizing the CR may be more important than minimizing the PRD. We have demonstrated that the tradeoff between CR and PRD varies based on biosignal and data compression method; thus, it is important to evaluate this tradeoff when evaluating data compression pipelines for each biosignal and each digital biomarker application.

Biomedical signal compression methods to reduce storage space with no loss of clinically significant information have focused on the elimination of redundant data in the signal [26]. The data compression methods utilized in this analysis were chosen based on their previously demonstrated ability to compress signals with similar characteristics to wearable sensor signals, including signals with varying redundancies, periodic behavior, and nonstationarity. Here, we have shown that different biosignals have varying performance with different compression methods, which may be dependent on signal characteristics, including redundancy and periodic behavior of the biosignals.

The tradeoff between CR and PRD should be carefully considered when compressing signals prior to digital biomarker development. Interestingly, while transformations prior to encoding increase the CR, they often also decrease the PRD. Therefore, if minimizing information loss is the priority, we suggest that researchers working with the data types analyzed here forego transformation and instead apply a direct-to-encoding method, for example, DDC with HE.

In this study, our secondary goal was to open-source algorithms and release data to advance biosignal compression methods development. The central repository containing biosignal compression methods that we have developed meets a critical need for community evaluation and development of data compression methods for wearable sensors as these data grow exponentially over time. In 2020, the total amount of digital healthcare data worldwide is projected to exceed 2000 exabytes (equivalent to 2 trillion GB) [2]; thus, data compression is a critical, timely need. We have shown with the Digital Biomarker Discovery Pipeline that open-sourcing tools, algorithms, and resources opens the door to innovation and helps establish best practices [10]. Open-source biosignal compression algorithms will improve validation practices, reduce duplication of research effort, and will accelerate innovation.

This study is limited in that we only utilized five biosignals (electrocardiogram, photoplethysmography, electrodermal activity, accelerometry, and skin temperature). Future studies should explore additional biosignals, including electroencephalographs and electromyograms. This study is limited in that we review only five combinations of data compression algorithms and encoders. However, by open-sourcing the Biosignal Data Compression Toolbox, we expect to cultivate a community where researchers will test and contribute novel data compression methods on the data that we have released, further expanding the reach of the current study. Recent research in using deep convolutional autoencoders for biosignal data compression [35,36] has shown great promise and should be included in the Biosignal Data Compression Toolbox in future work. Future work will involve developing and testing methods for quantifying signal characteristics to determine optimal data compression pipelines for other biosignals. Additionally, future work will include exploring wavelets for wavelet transform methods and optimizing these for various periodic biosignals. Future directions also include combining data compression methods with encryption methods in the Biosignal Data Compression Toolbox. Wearables data may be considered personal health information (PHI) and there are growing concerns over the level of deidentification that is possible for wearables data [37]; thus, combining encryption and data compression would help to mitigate privacy concerns.

This analysis and open-sourcing of biosignal data and compression methods serve as a launch point for the field of digital health to come together to contribute methods for wearable sensor data compression in the Biosignal Data Compression Toolbox. Information about contributing can be found at dbdp.org.

## Figures and Tables

**Figure 1 sensors-21-00516-f001:**
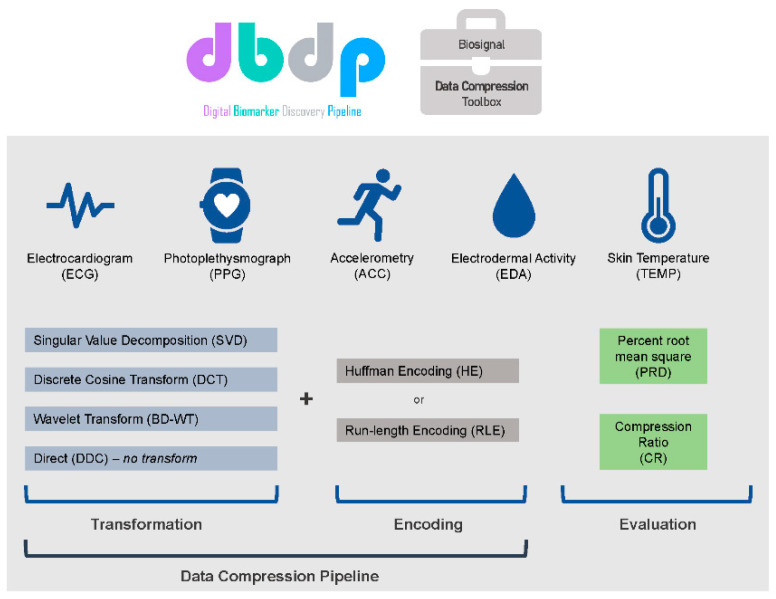
Graphical Abstract for the Biosignal Data Compression Toolbox.

**Figure 2 sensors-21-00516-f002:**
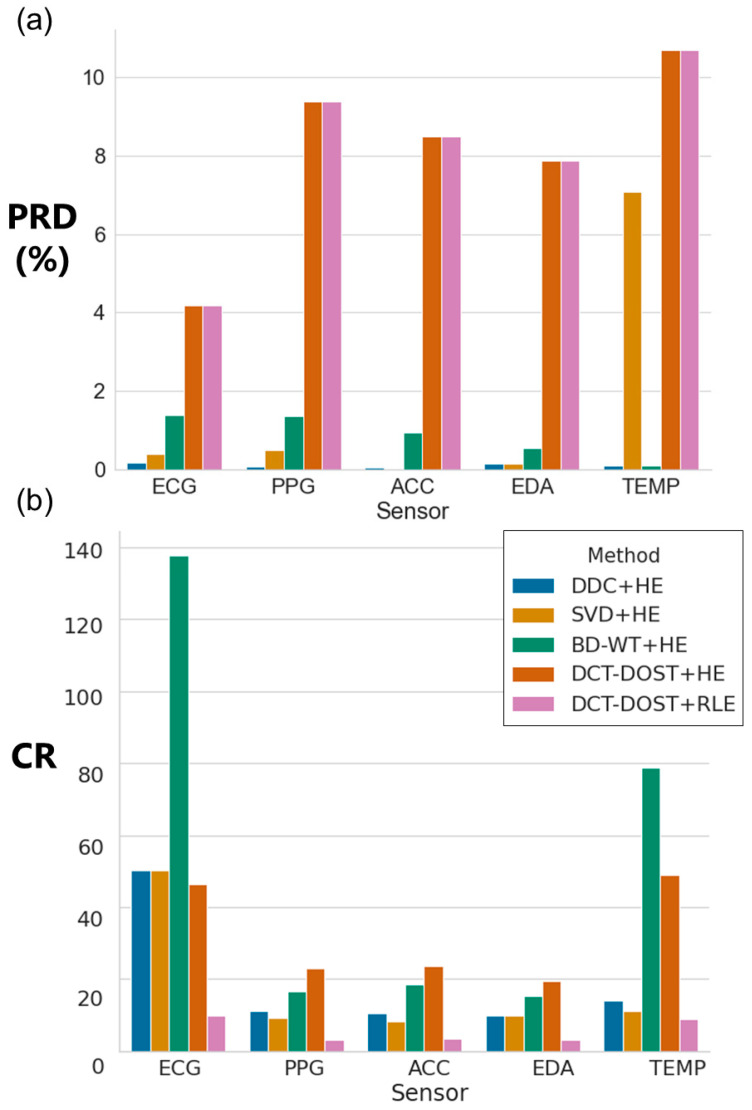
Comparison of (**a**) PRD and (**b**) CR for each method across the wearable sensors ECG, PPG, accelerometry (ACC), electrodermal activity (EDA), and skin temperature (TEMP).

**Figure 3 sensors-21-00516-f003:**
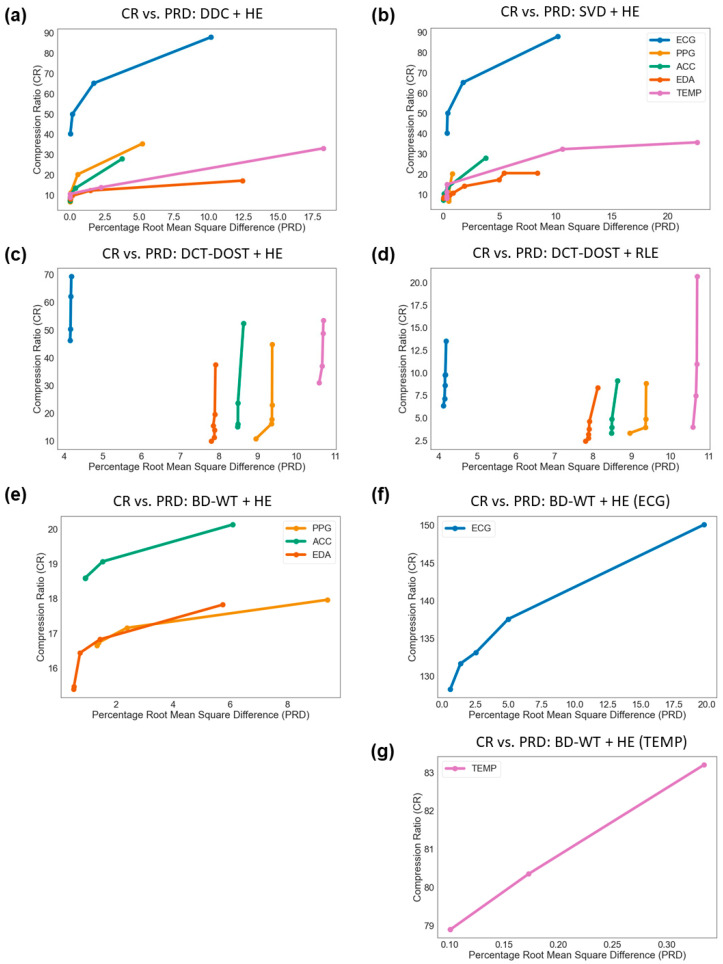
Compression ratio vs. PRD for each data compression pipeline evaluated: (**a**) Direct data compression (DDC) + Huffman encoding (HE), (**b**) singular value decomposition (*SVD*) *+ HE*, (**c**) discrete cosine transform (DCT)-based discrete orthogonal Stockwell transform (*DCT-DOST*) *+ HE*, (**d**) *DCT-DOST +* run-length encoding (*RLE*), (**e**–**g**) biorthogonal discrete wavelet transformation (*BD-WT) + HE*. The *BD-WT + HE* pipeline plot is broken down by sensor type due to the large-scale differences of the TEMP (**g**) and ECG (**f**) compression ratios.

**Figure 4 sensors-21-00516-f004:**
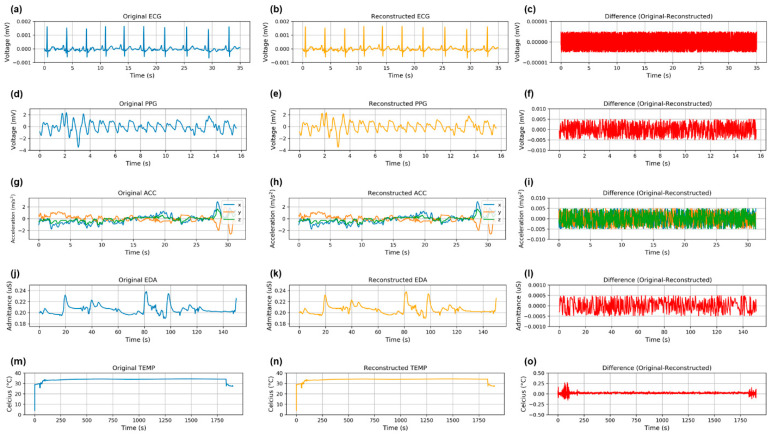
Comparison of original, reconstructed, and difference between original and reconstructed signals for ECG (**a**–**c**), PPG (**d**–**f**), ACC (**g**–**i**), EDA (**j**–**l**), and TEMP (**m**–**o**). These plots were constructed for the method that minimized the PRD: DDC with HE for ECG, PPG, ACC; *SVD with HE* for EDA; BD-WT with HE for TEMP. This figure enables visualization of the signal characteristics and corresponding differences between original and reconstructed signals.

**Table 1 sensors-21-00516-t001:** Compression Ratio (CR) and Percent Root-mean-square Difference (PRD%) of each data compression pipeline currently in the data compression toolbox for each of the wearable sensor biosignals: Electrocardiogram (ECG), photoplethysmography (PPG), accelerometry (ACC), electrodermal activity (EDA), and skin temperature (TEMP).

	ECG		PPG		ACC		EDA		TEMP	
	CR	PRD%	CR	PRD%	CR	PRD%	CR	PRD%	CR	PRD%
DDC + HE	50.16	0.17%	11.30	0.06%	10.40	0.04%	9.76	0.14%	13.99	0.09%
SVD + HE	50.16	0.38%	9.22	0.49%	8.63	0.02%	9.75	0.14%	11.32	7.08%
BD-WT + HE	131.70	1.37%	16.65	1.35%	18.59	0.94%	15.47	0.54%	78.90	0.10%
DCT-DOST + HE	46.32	4.18%	23.03	9.37%	23.75	8.49%	19.64	7.88%	48.82	10.69%
DCT-DOST + RLE	9.81	4.18%	3.30	9.37%	3.34	8.49%	3.19	7.88%	8.88	10.69%

## Data Availability

The data presented in this study will be available one month from the date of publication at dbdp.org.

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
