# Peer review of "Biosignal Compression Toolbox for Digital Biomarker Discovery"

_sensors, 2021, doi:10.3390/s21020516_

Round 1
Reviewer 1 Report
The paper was clearly written, it presented a practical approach for wearable data processing. It is better to incorporate more physiological data-set. Suggest to accept after minor revision.
Author Response
RESPONSE: Reviewer #1 suggests that the paper would be improved by including more physiological datasets. We agree that future studies that explore how the data compression methods that we developed here perform across many different types of biosignals would be interesting. In this study, we implement the data compression methods on five different biosignals that we collected- these are the five biosignals that are most commonly measured from wearable devices and therefore represent the key biosignals used for digital biomarker development. The purpose of this study was to demonstrate how our data compression methods perform for biosignals used for digital biomarker development. We achieve this goal with the data presented, and additional biosignals would both be out of scope and not change the conclusion of this paper. As such, we have updated the discussion to better explain the scope of the study and to include future directions for analysis of additional biosignals in lines 294-296:
“This study is limited in that we only utilized five biosignals (electrocardiogram, photoplethysmography, electrodermal activity, accelerometry, and skin temperature). Future studies should explore additional biosignals, including electroencephalograph and electromyogram.”
Reviewer 2 Report
Please see the attached file for comments.

Author Response
We have numbered the following responses based on the original comments made by Reviewer #2:
1.RESPONSE: Thank you for noting this. We revised the abstract to include all abbreviations (lines 20-21) and we checked the manuscript for abbreviations, correcting CR and PRD in line 186.
2.RESPONSE: We appreciate Reviewer #2’s recommendation to update the PRD equation to limit bias- this has greatly improved our manuscript. The PRD formula we previously used was based on older literature and in fact did not take into account the baseline mean of the dataset. We have updated the PRD definition based on the newer literature pointed out by Reviewer #2 and updated all tables and figures with this updated PRD definition, as detailed in the Methods (line 137), Results (lines 198,216-219, 233-237), Table 1, Figure 2, and Figure 3. With the updated PRD calculation, all of our recommendations were the same as the original manuscript with the exception that we recommend a different method for minimizing PRD in accelerometry, which we updated in the Abstract in line 25.
3.RESPONSE: We have added the requested recently published papers to the discussion in lines 297-299:
“Recent research in using deep convolutional autoencoders for biosignal data compression[30,31] has shown great promise and should be included in the Biosignal Data Compression Toolbox in future work.”
4. RESPONSE: Thank you for noting this discrepancy in our axis labels. We have updated Figure 4 as a result and we have thoroughly checked our other axes for any further discrepancies.
5.RESPONSE: We have extended the discussion of the applications for digital biomarkers in the introduction in lines 44-50 and 65-67 and provide more explanations on this application in the discission in lines 254-263:
Lines 44-50:
“Digital biomarkers have applications in a number of disease states, including movement-related disorders[6], breast cancer[7], and Alzheimer’s disease[8]. They are currently being used to diagnose and monitor a number of chronic diseases and conditions[9] and the field of digital biomarkers is rapidly expanding[10].”
Lines 254-263:
“In order to develop robust digital biomarkers, high resolution biosignal data is often necessary. However, the large volumes of longitudinal, high resolution data required for digital biomarker discovery accrue large storage costs. Healthcare data storage requirements are quadrupling every 2–3 years and are projected to cost up to $600 billion/month in 2020[3,5]. It is therefore critical to reduce biosignal data volumes. However, biosignal data must be preserved for digital biomarker discovery because digital biomarkers often require high precision. Thus, not only is it critical to reduce biosignal data volumes, but it is also necessary to ensure that information required for digital biomarker development is retained. The importance of robust data compression methods that have been tested on biosignal data is critical for developing digital biomarkers while minimizing data footprints.”
Reviewer 3 Report
Article is very well-written. There's a minor edit required.
Page 3 of 13: Place figure 1 after it was described in the text.
Author Response
RESPONSE: We have made this adjustment on page 3 to place the figure after its description in the text.
Round 2
Reviewer 2 Report
All my previous concerns have been responded to. No further comments were raised. Thus, the reviewer would like to suggest accepting this manuscript in the current form.